# Computer-guided design of optimal microbial consortia for immune system modulation

Richard R Stein[1,2,3,4†]*, Takeshi Tanoue[5,6†], Rose L Szabady[7], Shakti K Bhattarai[8], Bernat Olle[7], Jason M Norman[7], Wataru Suda[6,9], Kenshiro Oshima[9], Masahira Hattori[9], Georg K Gerber[10], Chris Sander[1,4,11], Kenya Honda[5,6], Vanni Bucci[8,12,13]*

[1]cBio Center, Department of Biostatistics and Computational Biology, Dana-Farber Cancer Institute, Boston, United States; [2]Department of Biostatistics, Harvard T.H. Chan School of Public Health, Boston, United States; [3]Department of Systems Biology, Harvard Medical School, Boston, United States; [4]Broad Institute of MIT and Harvard, Cambridge, United States; [5]RIKEN Center for Integrative Medical Sciences, Yokohama, Japan; [6]Department of Microbiology and Immunology, Keio University School of Medicine, Tokyo, Japan; [7]Vedanta Biosciences, Cambridge, United States; [8]Engineering and Applied Sciences PhD Program, University of Massachusetts Dartmouth, North Dartmouth, United States; [9]Graduate School of Frontier Sciences, The University of Tokyo, Kashiwa, Japan; [10]Massachusetts Host-Microbiome Center, Department of Pathology, Brigham and Women's Hospital, Harvard Medical School, Boston, United States; [11]Department of Cell Biology, Harvard Medical School, Boston, United States; [12]Department of Biology, University of Massachusetts Dartmouth, North Dartmouth, United States; [13]Center for Microbial Informatics and Statistics, University of Massachusetts Dartmouth, North Dartmouth, United States

**\*For correspondence:**
stein@jimmy.harvard.edu (RRS);
vanni.bucci@umassd.edu (VB)

†These authors contributed equally to this work

**Abstract** Manipulation of the gut microbiota holds great promise for the treatment of diseases. However, a major challenge is the identification of therapeutically potent microbial consortia that colonize the host effectively while maximizing immunologic outcome. Here, we propose a novel workflow to select optimal immune-inducing consortia from microbiome compositicon and immune effectors measurements. Using published and newly generated microbial and regulatory T-cell ($T_{reg}$) data from germ-free mice, we estimate the contributions of twelve Clostridia strains with known immune-modulating effect to $T_{reg}$ induction. Combining this with a longitudinal data-constrained ecological model, we predict the ability of every attainable and ecologically stable subconsortium in promoting $T_{reg}$ activation and rank them by the $T_{reg}$ Induction Score (TrIS). Experimental validation of selected consortia indicates a strong and statistically significant correlation between predicted TrIS and measured $T_{reg}$. We argue that computational indexes, such as the TrIS, are valuable tools for the systematic selection of immune-modulating bacteriotherapeutics.
DOI: https://doi.org/10.7554/eLife.30916.001

## Introduction

The intestinal microbiota has been shown to critically influence a multitude of host physiological functions, often through modulation of the immune system (*Gevers et al., 2014*; *Paun et al., 2016*).

**eLife digest** The role of the immune system is to protect the body from infection. It does this by using a powerful toolkit that isolates pathogens and removes damaged tissue. When directed against bacteria and viruses, the system helps to keep the body safe, but an imbalance of the components of the immune system can lead to inflammatory or allergic diseases.

The body has built-in mechanisms to shut off the immune response. For example, regulatory T-cells are immune cells with an anti-inflammatory effect, meaning they can switch off inflammation after the immune system has cleared an invasion. If their numbers are too low, it can contribute to unwanted inflammation. In ulcerative colitis, for instance, an unbalanced immune system mistakenly attacks healthy tissue. This causes inflammation and ulcers in the intestine, resulting in bleeding and weight loss.

Previous studies revealed that bacteria in the gut might help to control regulatory T-cell numbers. Certain combinations of bacteria can stimulate these regulatory T-cells and potentially dampen inflammation. Tweaking the different populations of microbes in the gut could provide a new way to treat diseases like ulcerative colitis. But, identifying the best combination to use is a major challenge. Testing them all one by one would be extremely challenging. Now, Stein, Tanoue et al. present a mathematical model designed to predict the best microbe-mix to use.

The model incorporated data from regulatory T-cells and microbes gathered from mice to estimate the contribution that different strains of bacteria make to regulatory T-cell numbers. This then fed into an ecological model predicted how different combinations of bacteria would behave in mice. The model aimed to fulfill two key criteria. First, to find combinations that can form stable, long-term bacterial colonies alone and when faced with competing microbes. Second, to boost regulatory T-cell numbers, helping them to expand to correct the immune imbalance.

To test the predictions, mice received combinations of bacteria suggested by the model. The model had predicted some combinations to be 'weak' at inducing regulatory T-cells, some 'intermediate' and some 'strong'. The results matched the predictions, validating the method.

This new model allows the rapid design of microbes that could dampen the immune response in the gut and paves the way for new treatments to correct imbalances of the immune system. By using already available data, it should be possible to expand the model to other diseases with imbalance in the immune system. In future, similar models could also find combinations for other medical uses. For example, to optimise the delivery of cancer immunotherapy, or to modulate the immune system after an organ transplant.

DOI: https://doi.org/10.7554/eLife.30916.002

Evidence includes studies in germ-free mice, which show that both pro-inflammatory T-helper 17 ($T_h17$) cells and anti-inflammatory regulatory T-cells ($T_{reg}$) are reduced in numbers in the intestinal lamina propria relative to conventional or specific pathogen-free mice (*Atarashi et al., 2008*), and that repopulation with specific bacteria can reconstitute them (*Atarashi et al., 2015*; *Ivanov et al., 2009*). One route to how the microbiota influences immune system activation is by the production of small molecules (*Donia and Fischbach, 2015*). Microbially produced short-chain fatty acids (SCFAs) were shown to facilitate extrathymic differentiation of immune system modulatory $T_{reg}$ (*Arpaia et al., 2013*; *Atarashi et al., 2013*; *Smith et al., 2013*) and are implicated in $T_{reg}$-dependent anti-inflammatory properties of a mix of 17 human-derived Clostridia strains (*Atarashi et al., 2013*; *Tanoue et al., 2016*). Interestingly, individual Clostridia strains only have a modest effect on $T_{reg}$ induction – optimal induction relies on the synergistic interplay of several strains (*Atarashi et al., 2013*; *Belkaid and Hand, 2014*). Because of these properties, there is currently great interest in the manipulation of this community for the treatment of inflammatory and allergic diseases (*Honda and Littman, 2012*; *Olle, 2013*; *Vieira et al., 2016*). Efforts involving transplantation of bacteria from healthy humans to humans with *C. difficile* infections or with metabolic diseases have provided evidence that microbiota repopulation could be used as possible strategy for disease prevention and treatment (*van Nood et al., 2013*; *Vrieze et al., 2012*). For this reason, intestinal supplementation of defined compositions of gut bacteria to treat a range of diseases is currently being pursued by multiple bio-pharmaceutical companies (*Garber, 2015*). Determining what microbial consortia can

colonize the host and stably coexist with an already resident microbial community, while inducing the desired immune response, is still a major challenge that hinders the translation of these efforts into the clinic (*Maldonado-Gómez et al., 2016*; *Weil and Hohmann, 2015*).

In previous work, some of us identified a set of 17 potent $T_{reg}$-inducing Clostridia strains to select candidate subsets from (*Atarashi et al., 2013*). The determination of maximally immune-phenotype inducing combinations from these 17 strains is, however, experimentally infeasible as it would require the testing of $2^{17}-1 = 131071$ possible subsets in mice (*Faith et al., 2014*). Therefore, a computational approach to prioritize which subsets to validate experimentally would have great utility. To our knowledge, to date there has been no model that allows for the simultaneous prediction of the dynamics of both the microbiota and the host immune response. Building on the approach to select for optimal bacterial combinations from (*Faith et al., 2014*) and other efforts aimed at identifying potential immune-modulating microbes (*Geva-Zatorsky et al., 2017*; *Schirmer et al., 2016*), we overcome this problem by proposing a mathematical modeling-based framework that, by resolving microbiome–immune system interactions with parameters constrained to experimental observations of microbiome and immune effectors, allows computational optimization of immune-stimulating bacterial combinations. To achieve this, we use a series of logically connected analyses that capture CD4$^+$FOXP3$^+$ $T_{reg}$ accumulation in the colonic lamina propria (*Omenetti and Pizarro, 2015*; *Round and Mazmanian, 2010*) in response to the dynamics of human-derived Clostridia strains in germ-free mice, which had previously been shown to potently induce $T_{reg}$ expansion (*Atarashi et al., 2013*) (*Figure 1*). To constrain this proposed model framework to experimental data, we combine previously published (*Atarashi et al., 2013*) and newly generated fluorescence-activated cell scanning (FACS) data with simulated and newly generated time-series colonization data from germ-free mice (*Figure 2A*). Subsequently, by applying a microbiome–$T_{reg}$ mathematical model to this combined data set, we infer each strain's individual contribution to the CD4$^+$FOXP3$^+$ $T_{reg}$ pool (*Figure 2B*) and obtain an estimate of every possible consortium's $T_{reg}$ induction potential. To justify the usage of predicted mono-colonization concentrations from a previously developed microbiome ecological model (*Bucci et al., 2016*), we validate its ability in predicting temporal dynamics in response to different subsets of $T_{reg}$-inducing strains (*Figure 3A*) and quantify the deviation of data and predictions (*Figure 3B*). Introduction of the TrIS, which assigns a score to every predicted steady-state microbial composition, enables us to identify combinations that robustly maximize $T_{reg}$ induction (*Figure 4A*). We analyze the relationship of TrIS and biomass (*Figure 4B*) as well as metabolic features of the consortia (*Figure 4C and D*). Most importantly, we demonstrate the utility of our approach in predicting the potency of selected microbial combinations by validating the $T_{reg}$ induction and colonization ability of model-predicted strong, intermediate and weak $T_{reg}$-inducing consortia *in vivo* (*Figure 4E*).

We envision that our framework, while in this study tailored to finding combinations ameliorating auto-inflammatory conditions, may also have direct relevance to other immune-system enhancing applications such as the optimal delivery of probiotic-based cancer immunotherapies (*Garrett, 2015*).

## Results

### Generation of multimodal microbiome–$T_{reg}$ data set

The goal of this study is to develop a mathematical modeling-based framework to rapidly and systematically select microbial consortia that maximize a desired immune outcome when introduced in a specific host microbial background. To achieve this, we combine (i) a microbiome ecological model that accurately describes the dynamics of these bacteria in the host with (ii) a microbiome–$T_{reg}$ mathematical model that characterizes the contribution of every strain to the immune phenotype of interest given corresponding microbiome colonization data (*Figure 1*). For (i), we gathered newly produced quantitative polymerase chain reaction (qPCR) colonization data from gnotobiotic mice and combined it with a previously published microbiome ecological model of the dynamics of 12 $T_{reg}$-inducing Clostridia strains that are part of an original consortium of 17 $T_{reg}$-inducing strains discovered by some of us (*Atarashi et al., 2013*) (see also *Table 1* for a breakdown of strains used in each study). In contrast to the original study of (*Bucci et al., 2016*), we included only 12 of the 13 strains used there because, based on the modeling and analysis of *Bucci et al. (2016)*, Strain 6 from

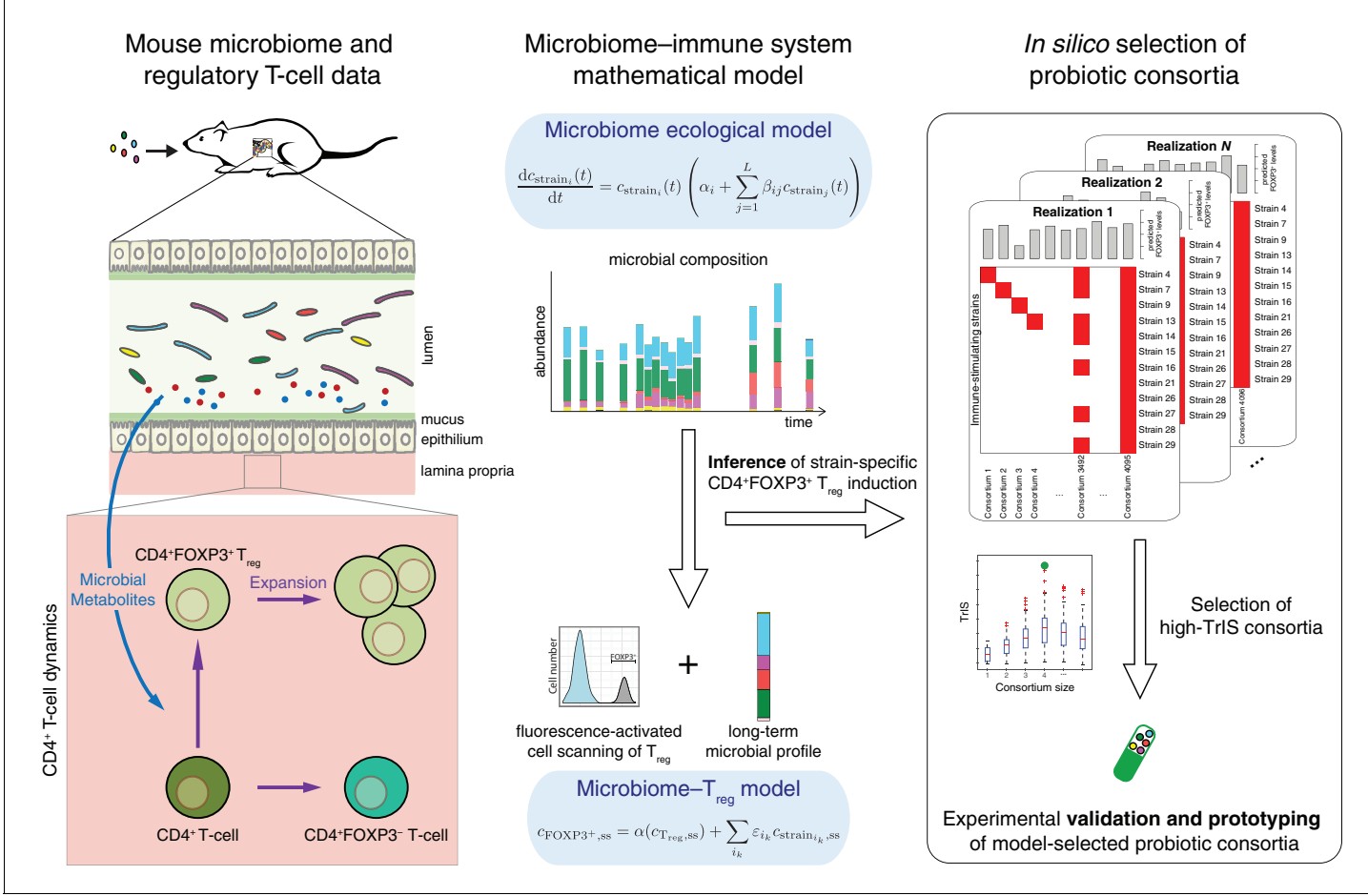

**Figure 1.** Conceptual figure. A microbiome–immune system mathematical model describing the activation of regulatory T-cells ($T_{reg}$) in response to colonization profiles of $T_{reg}$-stimulating Clostridia strains is the central element of this work. It consists of a previously derived microbiome ecological model that describes the short and long-term temporal dynamics of Clostridia strains in germ-free mice (*Bucci et al., 2016*) and is supplemented by a microbiome–$T_{reg}$ model of CD4$^+$FOXP3$^+$ $T_{reg}$ activation in response to long-term compositions in the microbiome. The individual contribution of each strain to the measured $T_{reg}$ activation is inferred using long-term colonization data from mouse experiments with subsets of these strains and corresponding measurements of $T_{reg}$ induction. The $T_{reg}$-induction score, TrIS, which accounts for ecological stability and immune activation, assigns a score to every possible strain combination and thereby identifies candidate probiotic consortia for experimental validation.

DOI: https://doi.org/10.7554/eLife.30916.003

the 13-strain set was predicted to not stably colonize in the presence of the other 12 strains. The published colonization data have been reported in our previous work (*Bucci et al., 2016*) and include time-series measurements of microbial abundances by qPCR under dietary perturbations. These are used to derive a predictive microbiome ecology model in gnotobiotic conditions based on an extension of the generalized Lotka–Volterra (gLV) equations (*Hofbauer and Sigmund, 1998*) as introduced in *Stein et al. (2013)*. For the newly generated dataset, we gavaged 14 mice with one of three possible 11-strain subsets from the 12-strain subset of original 13 strains (*Figure 2*) and used the derived stool measurements to validate the ability of our mathematical model in predicting unseen conditions. The three 11-strain subsets were chosen based on our 'keystoneness' definition, a measure describing the marginal predicted quantitative effect of removing each strain from the full community (*Bucci et al., 2016*). Specifically, we included two 11-strain combinations each missing one of the two highest keystoneness-scoring strains (VE202 Strain 15 and VE202 Strain 4) and one 11-strain combination which lacks the lowest keystoneness scoring strain (VE202 Strain 29). In analogy to *Bucci et al. (2016)*, each strain's density was profiled over time by qPCR with strain-specific primers (see Materials and methods). To resolve the contribution of each of these to $T_{reg}$ induction for point (ii) we coupled the colonization data from fecal content with newly collected and published

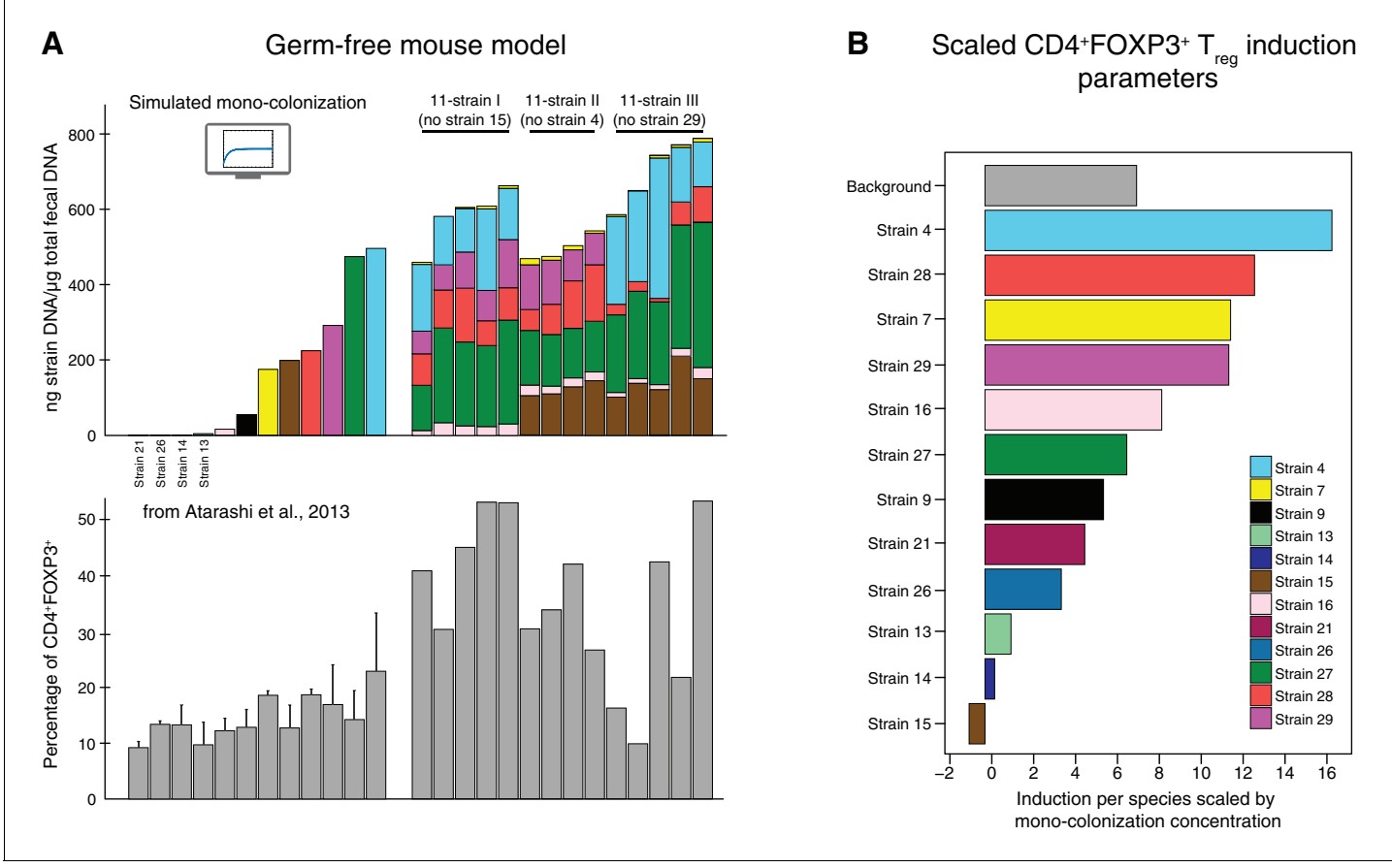

**Figure 2.** Data used for inference of the CD4+FOXP3+Treg induction parameters. (**A**) To infer the strain-resolved CD4+FOXP3+ Treg induction parameters CD4+FOXP3+ Treg abundance measurements and corresponding microbiome colonization data are used. CD4+FOXP3+ Treg data of single strains originates from previously published measurements from *Atarashi et al. (2013)*. Because in *Atarashi et al. (2013)* microbial mono-colonization levels were not measured, a previously published predictive model (*Bucci et al., 2016*) was used to simulate these experiments. In addition, newly generated CD4+FOXP3+ Treg and colonization mouse stool data from three 11-strain combinations was included into the analysis (see also *Figure 3*). The 11-strain combinations were chosen according to results of the 'keystoneness' analysis previously described in *Bucci et al. (2016)*. The microbiome compositions of these three 11-strain combinations were estimated by strain-specific qPCR. (**B**) The resulting CD4+FOXP3+ Treg induction parameters quantify the contribution of each individual strain to Treg induction. (Coefficients are scaled by the microbial mono-colonization concentrations for display reasons.)

DOI: https://doi.org/10.7554/eLife.30916.004

The following source data is available for figure 2:

**Source data 1.** Data containing the microbiome compositions, the corresponding measured CD4+FOXP3+ proportions and the derived induction parameters.
DOI: https://doi.org/10.7554/eLife.30916.005

FACS measurements of the CD4+FOXP3+ Treg population in the lamina propria of these mice (*Figure 2A*). As it is crucial to capture each strain's contribution alone and in combination with others, we also included CD4+FOXP3+ Treg measurements from our previously published mono-colonization experiments (*Atarashi et al., 2013*) (*Figure 2A*). Due to the fact that the single-strain mono-colonization concentrations were not measured in these experiments (*Atarashi et al., 2013*), we simulated them using the microbiome ecological model and parameters from (*Bucci et al., 2016*). This choice was supported by the model's capability in predicting unseen validation data (*Figure 3A*). Spearman's rank-order correlation coefficient ranges from 0.92 to 0.98 (p-value<$10^{-16}$) between observations and predictions depending on the time point (*Figure 3B*).

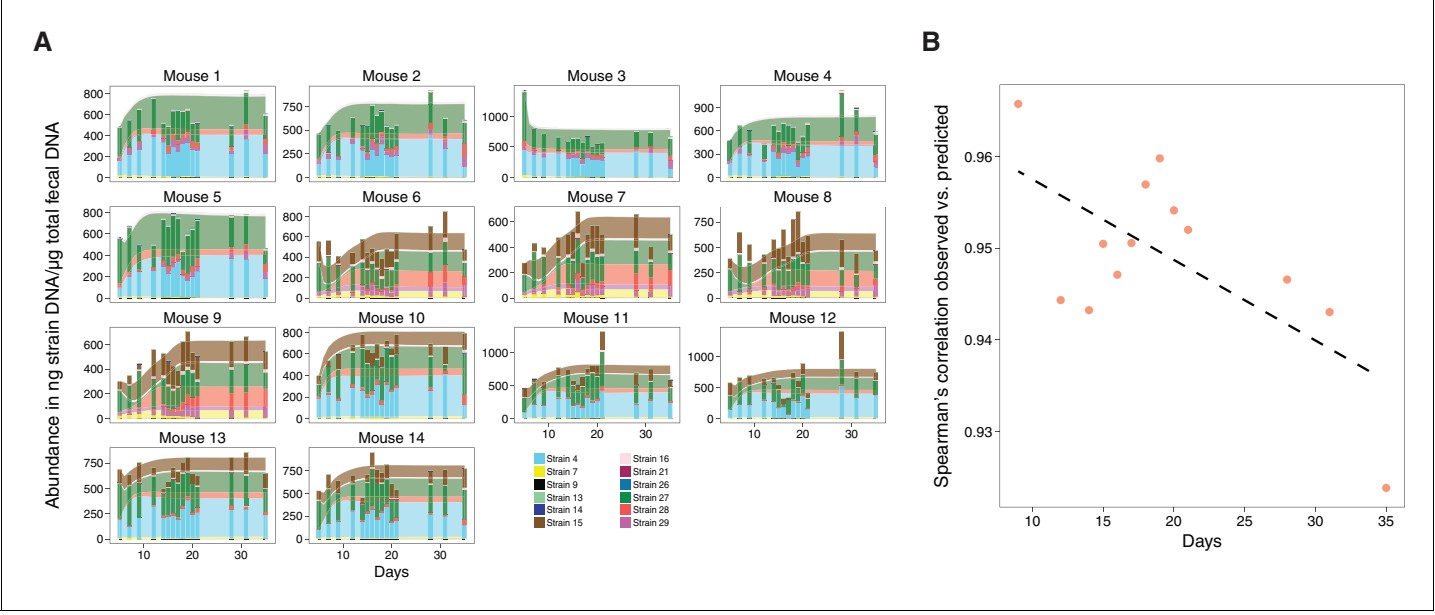

**Figure 3.** Validation of microbiome model describing the dynamics of Clostridia strains in germ-free mice. (**A**) The model-predicted dynamics of three 11-strain combinations from the 13 Clostridia strains described in *Bucci et al. (2016)* (areas) are compared to measured data from germ-free mice (stacked bars) and each panel corresponds to an individual mouse (*Figure 2A*). Predictions were obtained by numerically integrating the corresponding generalized Lotka–Volterra equations with parameters from *Bucci et al. (2016)* using only each mouse's initial microbial composition. In timelines/ mouse 1–5, Strain 15 is absent referring to 11-strain set I (five biological replicates); in timelines 6–9, Strain 4 is absent (11-strain set II; four biological replicates), and in timelines 10–14, Strain 29 is absent (11-strain set III; five biological replicates). Data were obtained by qPCR of genes specific to each strain. Densities are computed as averages of three technical replicates. The number of mice used in each condition was chosen consistently with previous experimental work (*Atarashi et al., 2013*; *Bucci et al., 2016*) and combined with extensive *in silico* testing of the inference error as a function of sampling frequency and noise levels (*Bucci et al., 2016*). (**B**) Spearman's rank correlation coefficient between observed and predicted data was computed at different time points. All displayed coefficients have a p-value of less than $10^{-16}$.

DOI: https://doi.org/10.7554/eLife.30916.006

## Derivation of the microbiome–$T_{reg}$ mathematical model

We used the described microbiome colonization data and corresponding CD4+FOXP3+ $T_{reg}$ measurements to determine the contribution of each strain to the $T_{reg}$ pool. We begin by subdividing the population of CD4+ T-cells into two major subpopulations depending on their intracellular FOXP3 expression: CD4+FOXP3+ $T_{reg}$ and the remainder among the CD4+ T-cells, the conventional CD4+-FOXP3− T-cells (*Bilate and Lafaille, 2012*; *Rudensky, 2011*). The concentration of CD4+ T-cells at time $t$ in the colonic lamina propria, $c_T(t)$, is then the combination of these two T-cell populations, $c_T(t) = c_{FOXP3+}(t) + c_{FOXP3-}(t)$. To include a variety of effects into our model, we assume T-cell dynamics to follow the gLV equations (*Gerber, 2014*; *Hofbauer and Sigmund, 1998*), which also account for the effect of the microbial strains in the lumen on the CD4+FOXP3− T-cell subpopulation. In addition, we use an extension of the standard gLV equations to include the impact of the Clostridia strains in terms of external perturbations (*Stein et al., 2013*). The resulting microbiome–$T_{reg}$ mathematical model is found as,

$$\frac{dc_{FOXP3+}(t)}{dt} = c_{FOXP3+}(t)\Bigg(\alpha_{FOXP3+} + \beta_{FOXP3+FOXP3+}c_{FOXP3+}(t)$$

$$+\beta_{FOXP3+FOXP3-}c_{FOXP3-}(t) + \sum_{k=1}^{K}\varepsilon_{i_k}c_{strain_{i_k}}(t)\Bigg)$$

(1)

where $\alpha_{FOXP3+}$ denotes the basal growth rate and $\beta_{FOXP3+FOXP3+}$ the self-interaction term of the CD4+FOXP3+ $T_{reg}$ population. The interaction terms $\beta_{FOXP3+FOXP3-}$ and $\beta_{FOXP3-FOXP3+}$ represent the effect of the CD4+FOXP3− T-cell on the CD4+FOXP3+ $T_{reg}$ population and of the CD4+FOXP3+ $T_{reg}$ on the CD4+FOXP3− T-cell population, respectively (*d'Onofrio, 2005*). Consequently, positive

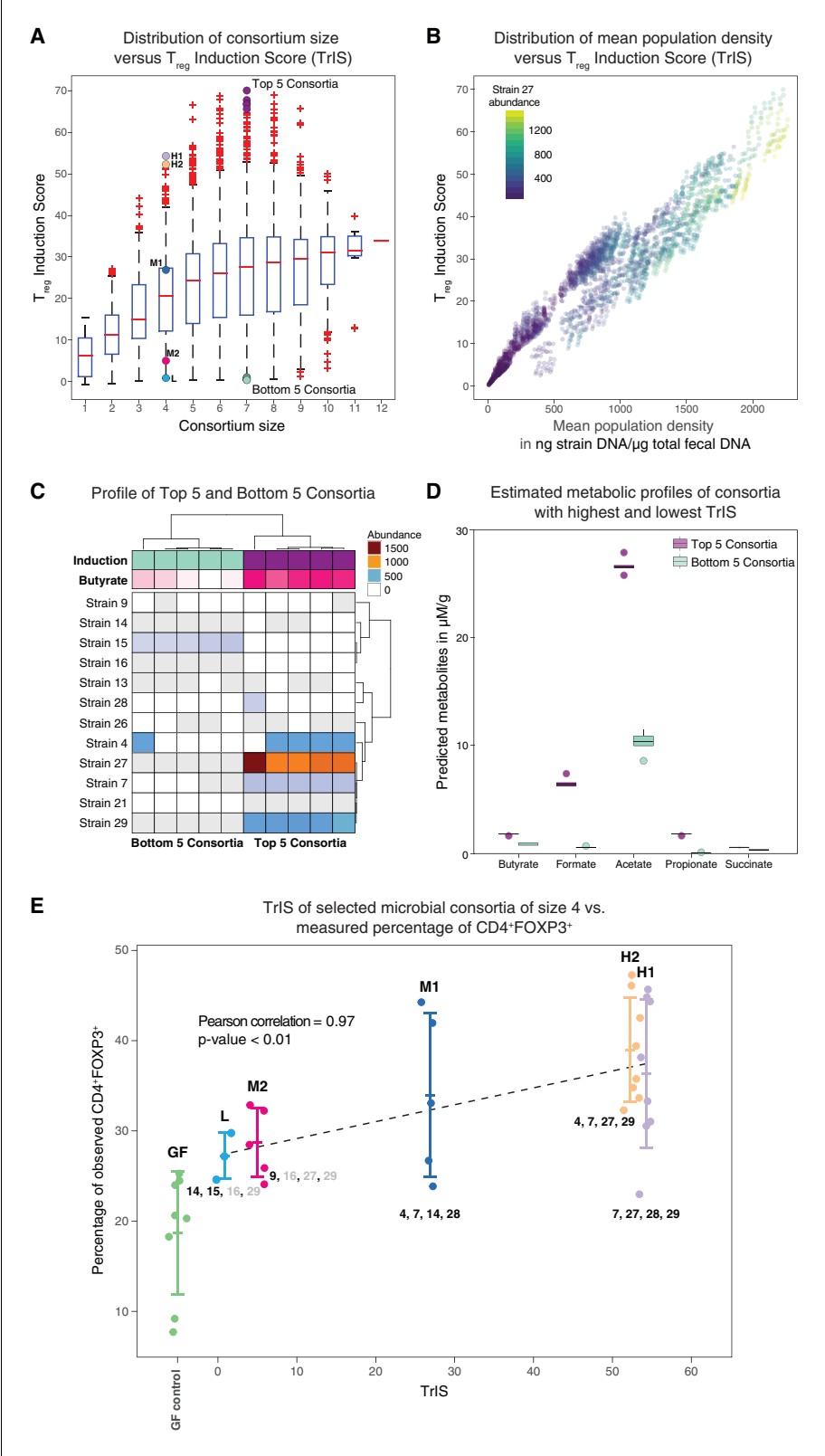

**Figure 4.** Identification and experimental validation of model-predicted $T_{reg}$-stimulating consortia. (**A**) Predicted $T_{reg}$ Induction Score (TrIS) in germ-free mice as a function of probiotic consortium size. Each data point represents one of the $2^N-1$ possible non-trivial consortia of size $N$. The five highest and lowest TrIS consortia of size seven are highlighted by violet and cyan filled dots, respectively, as well as, the 4-strain consortia used for experimental validation in subpanel E. (**B**) Distribution of the TrIS plotted against the total population density of each model-predicted consortium. This

*Figure 4 continued on next page*

*Figure 4 continued*

analysis shows a strong correlation (Spearman's Rank Correlation with coefficient 0.88 and p<0.05) between TrIS and total microbial density. The color of each dot represents the abundance of Strain 27, which is predicted to strongly colonize in high-TrIS subsets. (**C**) Heatmap of consortium membership of the five highest and lowest TrIS-consortia of size seven and their predicted T<sub>reg</sub> induction and estimated butyrate production using data from *Atarashi et al. (2013)*. (**D**) Estimation of potential SCFA output based on single-strain *in vivo* metabolic profiling from *Atarashi et al. (2013)*. The five highest TrIS consortia of size seven (from 4C) are predicted to have a significantly higher SCFA output relative to the five lowest ones. (**E**) Five 4-strain consortia were used for experimental validation: the two highest-ranked consortia (H1 and H2, respectively), the lowest-ranked consortium (L) and two intermediates (M1, M2). M1 and M2 were included because of interest in other disease areas. The experimentally introduced strains are listed next to each bar/consortium. Strains with numbers in black were detected by 16S rRNA sequencing in mouse stool samples, strains with numbers in gray were introduced but failed to colonize. A Pearson's correlation of 0.97 between the predicted TrIS and the average of each consortium's measured CD4⁺FOXP3⁺ T<sub>reg</sub> percentage proves the ability of the TrIS-based selection to correctly recover the experimentally observed T<sub>reg</sub> induction. (Pearson's correlation of all points has a value of 0.54.) Importantly, the H2 consortium displays an average increase in immune activity of 107% relative to the average germ-free mouse control. Eight biological replicates were used for GF, H1 and H2, five biological replicates were used for M1, M2 and three biological replicates were used for L. Replication and design of the validation experiment for T<sub>reg</sub> induction assessment is consistent with work from us and others (*Atarashi et al., 2013*; *Narushima et al., 2014*).

DOI: https://doi.org/10.7554/eLife.30916.007

The following source data and figure supplements are available for figure 4:

**Source data 1.** TrIS, TrIS rank and predicted median steady-state concentrations of the selected microbial consortia from *Figure 4E*.
DOI: https://doi.org/10.7554/eLife.30916.010

**Figure supplement 1.** TrIS versus estimated mean population density.
DOI: https://doi.org/10.7554/eLife.30916.008

**Figure supplement 2.** Metabolic profiling of three selected microbial consortia and comparison of predicted and measured metabolic profiles of three selected microbial consortia.
DOI: https://doi.org/10.7554/eLife.30916.009

interaction parameters correspond to activation, negative ones to inhibition. Moreover, $\varepsilon_{i_k}$ denotes the effect of strain $i_k$ on the CD4⁺FOXP3⁺ T<sub>reg</sub> population. For long-term observations, $t \to \infty$, the dynamics of $c_{\text{FOXP3}^+}(t)$ are given by its (non-trivial) steady-state solution, which simplifies the then static microbiome–T<sub>reg</sub> mathematical model of the relative CD4⁺FOXP3⁺ proportion in steady state, $r_{\text{FOXP3}^+,\text{ss}}$, to a linear regression problem,

$$r_{\text{FOXP3}^+,\text{ss}} = \tilde{\alpha} + \sum_{k=1}^{K} \tilde{\varepsilon}_{i_k} c_{\text{strain}_{i_k},\text{ss}}. \tag{2}$$

Here, we use that the absolute and relative abundance of the CD4⁺FOXP3⁺T<sub>reg</sub> population are coupled by $c_{\text{FOXP3}^{+/-},\text{ss}} = c_{\text{T,ss}} \cdot r_{\text{FOXP3}^{+/-},\text{ss}}$ (see Materials and methods). We assume that the steady-state CD4⁺ T-cell concentration is constant, $c_{\text{T,ss}} = \text{const.}$, across all microbial compositions. We justify this because we are dealing with genetically similar mice and a set of closely related Clostridia. This assumption is however not justifiable when comparing non-colonized and colonized germ-free mice (*Faith et al., 2014*).

Assigning to $r_{\text{FOXP3}^+,\text{ss}}$ the measured FACS-derived CD4⁺FOXP3⁺ T<sub>reg</sub> proportions after 35 days and to $c_{\text{strain}_{i_k},\text{ss}}$ the corresponding microbial profiles, we infer each strain's contribution to the CD4⁺-FOXP3⁺ T<sub>reg</sub> pool (*Figure 2B*) by solving *Equation (2)* with an $\ell^2$-penalized least-square regression with one shrinkage parameter, which is determined in a leave-one-sample-out cross-validation (*Stein et al., 2013*). The resulting normalized root-mean square deviation on left-out samples was found to be 12%.

## Derivation of the T<sub>reg</sub> Induction Score (TrIS) and selection of T<sub>reg</sub>-inducing consortia

After deriving a model to predict our candidate strains dynamics in germ-free conditions and having resolved each strain's contribution to T<sub>reg</sub> expansion, we aim to use this information to computationally select for consortia that maximize T<sub>reg</sub> induction while being ecologically robust (*Bucci et al., 2016*; *Stein et al., 2013*). To specify a measure of ecological robustness as well as immune induction potential for microbial consortia in germ-free mice, we define the T<sub>reg</sub> Induction Score (TrIS) as the average predicted regulatory T-cell activation of a given consortium of $K$ strains $\{\text{strain}_{i_1}, \cdots, \text{strain}_{i_K}\}$ while ignoring contributions from the host,

**Table 1.** List of strains used with respect to our previous published work.

The original publication (*Atarashi et al., 2013*) reported on 17 immune-modulating bacteria. In *Bucci et al. (2016)*, a dynamical eco-logical model for 13 of the original 17 strains was presented. These 13 strains were selected because they do not harbor antibiotic resistance genes. For our study, we used a 12-strain subset of these 13 strains based on ecological stability considerations. Coloniza-tion and CD4⁺FOXP3⁺T$_{reg}$ data for the three 11-strain sets, measured by qPCR and FACS, were used along with the simulated mono-colonization concentrations (*Figure 2*) and included into the microbiome–immune system model (*Figure 1*). The resulting parameters were employed to predict ecologically stable subsets with different T$_{reg}$ induction potentials. 4-strain subsets were selected to validate the predictions of the mathematical model (see main text and *Figure 3*). Strain 1: *Clostridium saccharogumia/Clostridium ramosum* JCM1298, Strain 3: *Flavonibacter plautii/Pseudoflavonifractor capillosus*, Strain 4: *Clostridium hathewayi/Clostridium saccharolyticum* WM1, Strain 6: *Blautia coccoides/Lachnospiraceae bacterium*, Strain 7: *Clostridium bolteae*, Strain 8: *Clostridium sp.* MLG055/*Erysipe-lotrichaceae bacterium 2 44A*, Strain 9: *Clostridium indolis/Anaerostipes caccae* DSM 14662, Strain 13: *Anaerotruncus colihominis*, Strain 14: *Ruminococcus sp.* ID8/*Lachnospiraceae bacterium 2 1 46FAA*, Strain 15: *Clostridium asparagiforme/Clostridium lavalense*, Strain 16: *Clostridium symbiosum*, Strain 18: *Clostridium ramosum*, Strain 21: *Eubacterium fissicatena/Eubacterium contortum/Clostrid-ium sp.* D5, Strain 26: *Clostridium scindens/Lachnospiraceae bacterium 5 1 57FAA*, Strain 27: *Lacnospiraceae bacterium A4/Lachno bacterium 3 1 57FAA CT1*, Strain 28: *Clostridium sp.* 316002/08, Strain 29: *Lacnospiraceae bacterium A4/Lachno bacterium 3 1 57FAA CT1*.

| | Atarashi et al., 2013 | Bucci et al., 2016 | 11-str. I | 11-str. II | 11-str. III | H1 | H2 | M1 | M2 | L |
|---|---|---|---|---|---|---|---|---|---|---|
| Strain 1 | x | | | | | | | | | |
| Strain 3 | x | | | | | | | | | |
| Strain 4 | x | x | x | | x | | | x | x | |
| Strain 6 | x | x | | | | | | | | |
| Strain 7 | x | x | x | x | x | x | x | x | | |
| Strain 8 | x | | | | | | | | | |
| Strain 9 | x | x | x | x | x | | | | x | |
| Strain 13 | x | x | x | x | x | | | | | |
| Strain 14 | x | x | x | x | x | | | x | | x |
| Strain 15 | x | x | | x | x | | | | | x |
| Strain 16 | x | x | x | x | x | | | | x | x |
| Strain 18 | x | | | | | | | | | |
| Strain 21 | x | x | x | x | x | | | | | |
| Strain 26 | x | x | x | x | x | | | | | |
| Strain 27 | x | x | x | x | x | x | x | | x | |
| Strain 28 | x | x | x | x | x | x | | x | | |
| Strain 29 | x | x | x | x | | x | x | | x | x |

DOI: https://doi.org/10.7554/eLife.30916.011

$$\text{TrIS}(\{\text{strain}_{i_1}, \cdots, \text{strain}_{i_K}\}) = \frac{1}{N} \sum_{n=1}^{N} \sum_{k=1}^{K} \tilde{\varepsilon}_{i_k} c^{(n)}_{\text{strain}_{i_k}, \text{ss}}. \tag{3}$$

If the predicted steady state of the microbial consortium $\left(c^{(n)}_{\text{strain}_{i_1}}, \ldots, c^{(n)}_{\text{strain}_{i_K}}\right)_{\text{ss}}$ that is computed from the $n$-th Markov Chain Monte Carlo (MCMC) parameter estimate (*Bucci et al., 2016*) is biologi-cally meaningful, i.e. positive, and stable, then $c^{(n)}_{\text{strain}_{i_k}, \text{ss}}$ denotes the steady-state concentration of strain $i_k$; otherwise $c^{(n)}_{\text{strain}_{i_k}, \text{ss}}$ is set to 0. Hence, the value of the TrIS is indicative of the expected CD4⁺FOXP3⁺ T$_{reg}$ induction (after removing the host contribution) and it is of the same unit as the FACS measurements. We evaluated TrIS for every possible strain combination that would stably col-onize the gut in germ-free background. In our computation, of the $2^{12}-1 = 4095$ possible steady-state strain configurations evaluated in $N = 22,500$ MCMC parameter estimates, 84% are found to be biologically meaningful and stable. Interestingly, while the average TrIS increases with consortium size, our analysis shows that a subset size of seven already contains bacterial combinations

maximizing induction (*Figure 4A*). Furthermore, in addition to the strong correlation between TrIS and the predicted total bacterial abundance in the consortium, we observed that high-induction consortia display an especially large enrichment in the abundance of Strain 27 (*Figure 4B*, *Figure 4—figure supplement 1*). Because short-chain fatty acids have been previously associated with colonic $T_{reg}$ induction (*Arpaia et al., 2013*) and increase in density upon supplementation with these strains (*Atarashi et al., 2013*), we decided to test if modeling-predicted high $T_{reg}$-inducing consortia were also enriched in SCFAs. We therefore compared the top 5-inducing microbial consortia of size seven against their same-size counterpart bottom 5 (*Figure 4C*). We predicted the SCFA concentration for each of the predicted compositions by summing the scaled metabolic outputs of each strain measured in mono-colonization experiments (*Atarashi et al., 2013*; *Narushima et al., 2014*) and normalized by the strain's model-predicted mono-colonization density. We performed a Welch two-samples t-test for each of the predicted SCFAs concentrations and found significant enrichment for all estimated SCFAs ($p < 0.05$, one tailed) in the high-TrIS consortia compared to the low ones (*Figure 4D*).

## Experimental validation of mathematical model predictions

We decided to experimentally test our approach's ability to correctly predict consortium ranking with respect to $T_{reg}$-induction. Due to regulatory constraints on the used probiotic strains – limiting us to a maximum of four strains at a time in follow-up experiments – we selected five 4-strain combinations with a variety of predicted immune effects. We measured $CD4^+FOXP3^+$ $T_{reg}$ induction for five microbial consortia of size four and the germ-free control. We chose the two highest TrIS consortia (H1, H2 with rank 1 and 2, respectively), the lowest one (L with rank 495) and two TrIS-intermediate consortia of interest (M1, M2 with rank 129 and 452, respectively). Strains contained in each of the five consortia are detailed in *Table 1*. We correlated the TrIS score with the mean observed $CD4^+FOXP3^+$ $T_{reg}$ percentage and found a significant Pearson correlation with coefficient of 0.97 and p-value<0.01 (*Figure 4E*). Importantly, when using 16S rRNA sequencing to investigate the resulting colonization profiles for these combinations, we observed that the high TrIS-scoring consortia (H1, H2, M1) all stably colonized while the two low-scoring consortia only displayed a subset of the introduced strains. This result remarkably reflects the nature of our scoring system which, in addition to immune activation potential, also incorporates colonization success (*Equation 3*).

Because the SCFA enrichment analysis from *in vivo* measurements of individual strains (*Figure 4C*) predicted acetate to be significantly increased in the five high versus low $T_{reg}$-inducing consortia of size seven, we decided to compare the measured acetic acid concentrations (*Figure 4—figure supplement 2A*) to predictions of acetate in the two high (H1 and H2) and low (L) $T_{reg}$-inducing 4-strain consortia (*Figure 4—figure supplement 2B*). ANOVA and subsequent post-hoc Tukey test showed significance in the measured enrichment of H1 compared to L (adjusted p-value<0.05) and of H1 compared to H2 (adjusted p-value<0.05) (*Figure 4—figure supplement 2A*). However, no statistical difference was observed between measurements in H2 and L (adjusted p-value<0.05). Remarkably, the model-predicted germ-free normalized acetate enrichment and the observed acetic acid concentration (normalized by the mean in germ-free conditions) are well correlated (*Figure 4—figure supplement 2B*).

## Discussion

Manipulation of the intestinal microbiota with defined bacterial consortia for the treatment of disease is a promising route for future therapeutics (*Hansen and Sartor, 2015*). However, choosing bacterial combinations from the vast combinatorial space of microbes that effectively colonize a host (possibly with a dysbiotic microbiota) and at the same time maximize a desired host phenotype requires an enormous number of expensive and time-consuming experimental trials (*Faith et al., 2014*). While data mining and statistical approaches could aid in this process, the majority of available microbiome analysis methods is still based on correlations (*Gerber, 2014*; *Morgan et al., 2015*) and inept at predicting unseen phenotypes. Identification of causal associations between microbes and host phenotype has been recently achieved by using an experimentally based microbe–phenotype triangulation (*Surana and Kasper, 2017*). However, this approach remains impractical when the goal is to explore and rank all possible microbiome consortium combinations with respect to a host phenotype of interest. In this study, we leveraged our previous computational

methods to forecast temporal microbiome dynamics and make predictions on stability in the context of infectious and inflammatory diseases including *C. difficile* colonization and inflammatory bowel disease (*Bucci et al., 2016*; *Buffie et al., 2015*; *Stein et al., 2013*).

Specifically, we have presented a novel modeling-based method that combines dynamical predictions from data-driven models of the microbiome and the interactions between microbes and the immune response (such as CD4$^+$FOXP3$^+$ T$_{reg}$) and thereby enables the rational design of immune-modulating bacterial consortia. Limiting our scope to long-term steady-state dynamics, we were able to derive a microbiome–T$_{reg}$ mathematical model which is constrained by experimental observations from multimodal data. In the presented work, the used data modalities are qPCR and 16S rRNA sequencing data to estimate microbial abundances and FACS to assess CD4$^+$FOXP3$^+$ T$_{reg}$ induction, respectively. However, the proposed framework is naturally extensible to other host data, for example data from metabolite profiling, immune readouts (e.g. CD8 T-cell activation, T$_h$1/T$_h$17 cell depletion) or host transcriptome profiling (*Atarashi et al., 2017*; *Morgan et al., 2015*). To evaluate candidate consortia that could be relevant to the treatment of auto-inflammatory diseases, we introduced the T$_{reg}$ Induction Score, TrIS, as a novel metric that accounts for both ecological stability and efficacy in immune modulation. This metric allowed us to rationally identify combinations that would stably colonize and at the same time produce substantial T$_{reg}$ induction in germ-free mice. Remarkably, in validation experiments of five distinct modeling-guided consortia with predicted diverse induction potential, we proved the ability of our approach to successfully select microbial combinations with a desired therapeutic activity. To our knowledge, this is to date, the first study in which observation-constrained *in silico* modeling of microbiome and host phenotype has successfully guided the rational design of drugs of defined bacterial consortia.

Our work relies on the gLV model assumption for both microbiome and CD4$^+$ regulatory T-cell dynamics which, despite being very versatile, has some limitations including the lack of third or higher order interactions or saturation effects (*Wangersky, 1978*). Moreover, with the data available to us, we needed to assume that the overall T-cell density at steady state is constant across mice and different microbiome compositions. While this assumption was reasonable for the data we analyzed (see Results), the addition of future measurements on total T-cell abundance will likely improve the prediction accuracy of our model as well as provide important insight into the variability of T-cell population size in response to microbial immune induction.

We performed our data and modeling analysis in germ-free mice. While previous work by some of us showed that the amount of T$_{reg}$ induction is independent of the germ-free background (IQI, Balb/c or C57BL/6) (*Atarashi et al., 2015*), it is noteworthy that our model, which was trained on IQI mice data, robustly predicted both T$_{reg}$ induction and acetate enrichment of C57BL/6 colonized mice as used in the validation experiments.

Our predictions suggest some degree of consistency in terms of functional outcome (e.g. enrichment in acetate) of high-scoring consortia relative to low-scoring ones in germ-free mice. However, before translating these findings into therapeutic development, future studies will need to be performed in not germ-free settings (e.g. SPF mice, humanized mice) in order to account for the effects of an already-established flora. Individual-specific microbiome features may constrain the colonization potential of the selected strains due to specific ecological network effects (*Smits et al., 2016*), which suggests the need for a careful characterization of the ecological interactions between the proposed probiotic product and a specific recipient community (e.g. a single ulcerative colitis patient (*Atarashi et al., 2013*)).

Treatment of auto-immune diseases overall is not necessarily achieved by the exclusive optimization of one objective function (e.g. for ulcerative colitis, the maximization of T$_{reg}$ activation), but may need the simultaneous manipulation of a multi-process host immune spectrum, which could include the concomitant reduction of pro-inflammatory phenotypes (*Atarashi et al., 2015*). Given new data availability and careful experimental design, we believe that the modeling framework proposed in this study could overcome these problems by introducing constraints into the scoring metric that account for pre-colonized mouse model features. To the limit, in the presence of a large training data set that would include sufficient information on individual patient variation, our workflow will give us the unprecedented potential of testing microbiota manipulation on a personalized level *in silico*.

Mathematical modeling-based methods have the potential to greatly accelerate the development of treatment of human disease (*Michor and Beal, 2015*). In this work, we develop and demonstrate in a first-of-a-kind experimental validation the usefulness of a mathematical microbiome–immune

system model. It can be considered a stepping-stone to the accelerated prototyping and rational design of microbiome therapies.

# Materials and methods

## Derivation of the microbiome–T$_{reg}$ induction mathematical model

We assume the following dynamics for the CD4$^+$FOXP3$^+$ regulatory T-cell population:

$$\frac{dc_{\text{FOXP3}^+}(t)}{dt} = c_{\text{FOXP3}^+}(t) \left( \alpha_{\text{FOXP3}^+} + \beta_{\text{FOXP3}^+\text{FOXP3}^+} c_{\text{FOXP3}^+}(t) + \beta_{\text{FOXP3}^+\text{FOXP3}^-} c_{\text{FOXP3}^-}(t) + \sum_{k=1}^{K} \varepsilon_{i_k} c_{\text{strain}_{i_k}}(t) \right).$$

Here, $\alpha_{\text{FOXP3}^+}$ denotes the basal growth rate and $\beta_{\text{FOXP3}^+\text{FOXP3}^+}$ the self-interaction term of the CD4$^+$FOXP3$^+$ T$_{reg}$ population, while the interaction parameters $\beta_{\text{FOXP3}^+\text{FOXP3}^-}$ and $\beta_{\text{FOXP3}^-\text{FOXP3}^+}$ characterize the effect of the CD4$^+$FOXP3$^-$ T-cells on the CD4$^+$FOXP3$^+$ T$_{reg}$ population and of the CD4$^+$FOXP3$^+$ T$_{reg}$ on the CD4$^+$FOXP3$^-$ T-cell population, respectively (*d'Onofrio, 2005*). Moreover, $\varepsilon_{i_k}$ denotes the effect of strain $i_k$ on the CD4$^+$FOXP3$^+$ T$_{reg}$ population. The non-trivial steady-state solution (i.e., the algebraic solution of the right-hand side of *Equation (1)* set to 0 with $c_{\text{FOXP3}^+} \neq 0$) is found as,

$$c_{\text{FOXP3}^+,\text{ss}} = -\frac{1}{\beta_{\text{FOXP3}^+\text{FOXP3}^+}} \left( \alpha_{\text{FOXP3}^+} + \beta_{\text{FOXP3}^+\text{FOXP3}^-} c_{\text{FOXP3}^-,\text{ss}} + \sum_{k=1}^{K} \varepsilon_{i_k} c_{\text{strain}_{i_k},\text{ss}} \right).$$

Using $c_{\text{T,ss}} = c_{\text{FOXP3}^+,\text{ss}} + c_{\text{FOXP3}^-,\text{ss}}$, the steady-state concentrations of the CD4$^+$FOXP3$^+$ T$_{reg}$ and CD4+FOXP3$^-$ populations, $c_{\text{FOXP3}^{+/-},\text{ss}}$, are derived from the FACS-based relative abundances $r_{\text{FOXP3}^{+/-},\text{ss}}$ by, $c_{\text{FOXP3}^{+/-},\text{ss}} = c_{\text{T,ss}} \cdot r_{\text{FOXP3}^{+/-},\text{ss}} = c_{\text{T,ss}}(1 - r_{\text{FOXP3}^{-/+},\text{ss}})$. Finally, the linear relationship between the relative abundances, $r_{\text{FOXP3}^+,\text{ss}}$, and the strain densities, $c_{\text{strain}_{i_k},\text{ss}}$, is found as,

$$r_{\text{FOXP3}^+,\text{ss}} = \frac{1}{\beta_{\text{FOXP3}^+\text{FOXP3}^-} - \beta_{\text{FOXP3}^+\text{FOXP3}^+}} \left( \frac{\alpha_{\text{FOXP3}^+}}{c_{\text{T,ss}}} + \beta_{\text{FOXP3}^+\text{FOXP3}^-} + \frac{1}{c_{\text{T,ss}}} \sum_{k=1}^{K} \varepsilon_{i_k} c_{\text{strain}_{i_k},\text{ss}} \right)$$

$$\equiv \tilde{\alpha} + \sum_{k=1}^{K} \tilde{\varepsilon}_{i_k} c_{\text{strain}_{i_k},\text{ss}}$$

assuming constant concentration of CD4$^+$ T-cells, $c_{\text{T,ss}} = \text{const.}$, across all possible microbiome compositions. The unknown parameters $\tilde{\alpha}$ and $\tilde{\varepsilon}_{i_k}$ are estimated in an $\ell^2$-penalized least-square regression (so-called ridge regression) with a positive shrinkage parameter $\lambda$, which is determined in a leave-one-sample-out cross-validation as $\lambda^* = 2$.

## Collection of experimental data for training the microbiome–immune system model

### Strain abundance profiling

In our previous work (*Bucci et al., 2016*), we have inferred a mathematical model describing the dynamics of a 13-strain subset from the original 17-strain human-derived Clostridia consortium from *Atarashi et al. (2013)* in germ-free mice. Using newly generated data from the same experimental setup, we are now able to assess its predictive quality for three distinct 11-strain subsets. The three 11-strain compositions were selected based on their capability to maintain stability in the simulations when either one of two high (Strain 15 and Strain 4, here referred to as cases I and II, respectively) or one low (Strain 29; III) keystone strain was removed (*Bucci et al., 2016*). For the experimental validation, germ-free IQI mice were purchased from Sankyo Laboratories (Japan), randomized and maintained in germ-free vinyl isolators in the animal facility of RIKEN. Twelve T$_{reg}$-inducing Clostridia strains were selected from the previously reported VE202 consortium consisting of 17 T$_{reg}$-inducing strains (4, 7, 9, 13–16, 21, 26–29) and were individually cultured in modified Eggerth Gagnon broth under strictly anaerobic conditions (80% N$_2$, 10% H$_2$, 10% CO$_2$) at 37°C in an anaerobic chamber (Coy Laboratory Products, Grass Lake, MI) to confluence. The cultured bacterial strains were then mixed and the three mixtures of 11 strains (described above) were orally inoculated into five IQI germ-free adult mice each. One mouse for condition II died and was therefore discarded from the

study. After an initial 9-day interval of acclimation, we collected fecal pellets at 2–4 days interval until day 35, time at which mice were euthanized and analyzed for CD4$^+$FOXP3 induction as in *Atarashi et al. (2013)*. Colonization levels for each strain were assessed by amplifying strain-specific regions with qPCR as described in *Bucci et al. (2016)*. Bacterial genomic DNA was extracted from 1 to 2 fecal pellets using QIAamp DNA Stool Mini Kit (Qiagen, Hilden, Germany). The amount of DNA was quantified using a Qubit dsDNA HS assay kit and Qubit fluorometer (Invitrogen, Carlsbad, CA). DNA was then subjected to qPCR using Thunderbird SYBR qPCR Mix (TOYOBO, Osaka, Japan) and a LightCycler 480 (Roche) with primers specific to 16S ribosomal RNA (rRNA) genes of the 12 Clostridia strains as in *Bucci et al. (2016)*. Quantification of each strain in each sample was accomplished using standard curves of known concentrations of DNAs purified from each strain individually cultured *in vitro*. Strain densities in each sample were calculated by dividing the above absolute quantification numbers by the weight of the extracted fecal DNA. The 11-strain experiments were ethically approved by RIKEN, Keio and Azabu Universities under protocol H24-9(14) from RIKEN.

## Estimation of CD4$^+$FOXP3$^+$ T$_{reg}$ – Isolation of intestinal lamina propria lymphocytes and flow cytometry

The colons were collected and opened longitudinally, washed with PBS to remove all luminal contents and shaken in Hanks' balanced salt solution (HBSS) containing 5 mM EDTA for 20 min at 37°C. After removing epithelial cells, muscle layers and fat tissue using forceps, the lamina propria layers were cut into small pieces and incubated with RPMI1640 containing 4% fetal bovine serum, 0.5 mg/ml collagenase D, 0.5 mg/ml dispase and 40 mg/ml DNase I (all Roche Diagnostics, Risch-Rotkreuz, Switzerland) for 1 hr at 37°C in a shaking water bath. The digested tissues were washed with HBSS containing 5 mM EDTA, resuspended in 5 ml of 40% Percoll (GE Healthcare, Boston, MA) and overlaid on 2.5 ml of 80% Percoll in a 15 ml Falcon tube. Percoll gradient separation was performed by centrifugation at 850 $g$ for 25 min at 25°C. The lamina propria lymphocytes were collected from the interface of the Percoll gradient and suspended in ice-cold PBS. For analysis of T$_{reg}$, isolated lymphocytes were labeled with the LIVE/DEAD fixable dead cell stain kit (Life Technologies, Carlsbad, CA) to exclude dead cells from the analysis. Then, surface and intracellular staining of CD3, CD4 and FOXP3 was performed using the BV605-labeled anti-CD3 (17A2, Biolegend, San Diego, CA), BV421-labeled anti-CD4 (RM4-5, Biolegend), Alexa700-labeled anti-FOXP3 antibody (FJK-16 s, eBioscience, San Diego, CA), and FOXP3 staining buffer set (eBioscience). The antibody-stained cells were analyzed with LSR Fortessa and data were analyzed using FlowJo software (Tree Star, Ashland, OR).

## Measurement of organic acids

Organic acid concentrations in caecal contents were determined by gas chromatography-mass spectrometry (GC-MS). Caecal contents (10 mg) were disrupted using 3 mm zirconia/silica beads (BioSpec Products) and homogenized in extraction solution containing 100 ml of internal standard (100 mM crotonic acid), 50 ml of HCl and 200 ml of ether. After vigorous shaking using a Shakemaster neo (Bio Medical Science) at 1500 rpm for 10 min, homogenates were centrifuged at 1000 $g$ for 10 min and then the top ether layer was collected and transferred into new glass vials. Aliquots (80 ml) of the ether extracts were mixed with 16 ml of N-tert-butyldimethylsilyl-N-methyltrifluoroacetamide (MTBSTFA). The vials were sealed tightly by screwing and heated at 80°C for 20 min in a water bath, and left at room temperature for 48 hr for derivatization. The samples were then run through a 6890N Network GC System (Agilent Technologies) equipped with HP-5MS column (0.25 mm 330 m 30.25 mm) and 5973 Network Mass Selective Detector (Agilent Technologies, Santa Clara, CA). Pure helium (99.9999%) was used as a carrier gas and delivered at a flow rate of 1.2 ml/min. The head pressure was set at 10 psi with split 10:1. The inlet and transfer line temperatures were 250 µC and 260 µC, respectively. The following temperature program was used: 60 µC (3 min), 60–120°C (5°C/min), 120–300°C (20°C/min). One microliter quantity of each sample was injected with a run time of 30 min. Organic acid concentrations were quantified by comparing their peak areas with the standards.

## Numerical simulations of the three 11-strain subsets used for microbiome–T$_{reg}$ model training

We used 22,500 sets of Markov Chain Monte Carlo generalized Lotka–Volterra parameter sets determined by applying the Bayesian Variable Selection algorithm within MDSINE to the data of

*Bucci et al. (2016)*, and the first time point of measured microbial profiles for each of the 14 validation mice as initial condition, to simulate the gLV system of differential equations corresponding to each mouse microbiome (*Figure 3A*). Prediction accuracy was evaluated by calculating the Spearman correlation coefficient between observed and predicted data (*Figure 3B*).

## Simulation of mono-colonization abundances

As the experimental data from *Atarashi et al. (2013)* only provided $CD4^+FOXP3^+$ T-cell levels for the mono-strain colonization experiments at 35 days after inoculation but no measurement of the long-term microbial concentrations in the gut, we used instead the corresponding estimated mono-strain colonization densities obtained from the gLV model (section above and *Figure 2A*). In general, the long-term behavior of the gLV system is determined by its steady states which are uniquely defined by the inferred model parameters (*Stein et al., 2013*). We computed the parameter median in each single model variable from the 22,500 MCMC parameter sets. These median parameters were then used to deduce the steady-state densities, which were together with the measured $CD4^+$-FOXP3 proportions included into the training of the microbiome–$T_{reg}$ model.

## Collection of $CD4^+FOXP3^+$ $T_{reg}$ data from 4-strain experiments for validation of the modeling-based predictions

Bacterial strains 4, 7, 9, 14, 15, 16, 27, 28, 29 were grown anaerobically in PYG broth (Peptone, Yeast and Glucose broth from Anaerobe Systems, Cat no: AS-822) until they reached stationary phase (48 hr for strains 27 and 29, 24 hr for the remaining strains). Each 200 µl-mouse dose of a 4-strain LBP contained 50 µl of 20 times concentrated stationary phase culture. Germ-free C57BL/6 mice aged 6–8 weeks were randomized and gavaged with a total dose of $5 \cdot 10^7$–$2 \cdot 10^8$ bacteria in a 200 µl, and maintained under gnotobiotic conditions for four weeks. Use of a C57BL/6 background for these experiments was motivated by availability of animals at the facility where we performed the validation and justified by the fact that previous work from us has shown that $T_{reg}$ induction by our Clostridia strains does not differ between Balb/c, IQI, and C57BL/6 mice (*Atarashi et al., 2015*). Mice were then sacrificed, colons harvested, and lamina propria leukocytes isolated and stained for $CD3^+CD4^+FOXP3^+$ $T_{reg}$ as described above. Eight mice each were used for consortia High 1 (H1-strains: 7, 27, 28, 29) and High 2 (H2-strains: 4, 7, 27, 29). Five mice each were used for the intermediate high (M1-strains: 4, 7, 14, 28), and intermediate low consortia (M2-strains: 9, 16, 27, 29). Three mice were used for Low 1 (L1-strains: 14, 15, 16, 29). Colonization profiling was determined through 16S rRNA sequencing (as above) and verified by blasting representative sequences to a 16S VE202 fasta database. The 4-strain validation experiments were performed in the Massachusetts Host Microbiome Center under IACUC protocol 2016N000141.

# Additional information

## Competing interests

Takeshi Tanoue, Vanni Bucci: Has received support from Vedanta Biosciences, Inc. under research agreement with his institution. Rose L Szabady, Jason M Norman: Is employee of Vedanta Biosciences, Inc. Bernat Olle: Is the Chief Executive Officer of Vedanta Biosciences, Inc. Georg K Gerber: Is a member of the Scientific Advisory Board of Kaleido, Inc. Kenya Honda: Is a Co-Founder and Scientific Advisory Board Member of Vedanta Biosciences, Inc. Has received support from Vedanta Biosciences, Inc. under research agreements with his institution. The other authors declare that no competing interests exist.

## Funding

| Funder | Grant reference number | Author |
| --- | --- | --- |
| National Institutes of Health | P41 GM103504 | Richard R Stein<br>Chris Sander |
| Defense Advanced Research Projects Agency | BRICS award HR0011-15-C-0094 | Georg K Gerber |

| Human Frontier Science Program | RGP00055/2015 | Chris Sander |
| --- | --- | --- |
| Takeda Science Foundation | | Kenya Honda |
| National Institute of General Medical Sciences | 5R01 GM106303 | Chris Sander |
| Japan Agency for Medical Research and Development | | Kenya Honda |
| National Institute of Allergy and Infectious Diseases | R15-AI112985-01A1 | Vanni Bucci |
| National Science Foundation | 1458347 | Vanni Bucci |
| Core Research for Evolutional Science and Technology | | Kenya Honda |
| Brigham and Women's Hospital | Precision Medicine Initiative | Georg K Gerber |

The funders had no role in study design, data collection and interpretation, or the decision to submit the work for publication.

## Author contributions

Richard R Stein, Vanni Bucci, Conceptualization, Data curation, Software, Formal analysis, Supervision, Investigation, Visualization, Methodology, Writing—original draft, Writing—review and editing; Takeshi Tanoue, Formal analysis, Validation, Investigation; Rose L Szabady, Data curation, Validation, Investigation; Shakti K Bhattarai, Jason M Norman, Wataru Suda, Kenshiro Oshima, Masahira Hattori, Formal analysis; Bernat Olle, Georg K Gerber, Chris Sander, Conceptualization; Kenya Honda, Conceptualization, Formal analysis

## Author ORCIDs

Richard R Stein  http://orcid.org/0000-0002-5110-6863
Georg K Gerber  http://orcid.org/0000-0002-9149-5509
Vanni Bucci  http://orcid.org/0000-0002-3257-2922

## Ethics

Animal experimentation: 11-strain time-series mouse experiments were performed under ethical approval by RIKEN, Keio and Azabu Universities under protocol H24-9(14) (RIKEN). 4-strain validation mouse work was performed at Brigham and Women's Hospital in Boston, MA in the Massachusetts Host Microbiome Center under IACUC protocol 2016N000141.

## Decision letter and Author response

Decision letter https://doi.org/10.7554/eLife.30916.016
Author response https://doi.org/10.7554/eLife.30916.017

# Additional files

## Supplementary files

• Transparent reporting form
DOI: https://doi.org/10.7554/eLife.30916.012

## Major datasets

The following previously published dataset was used:

| Author(s) | Year | Dataset title | Dataset URL | Database, license, and accessibility information |
| --- | --- | --- | --- | --- |
| Atarashi K, Tanoue T, Ando M, Kamada N, Nagano Y, | 2015 | Th17 Cell Induction by Adhesion of Microbes to Intestinal Epithelial Cells | http://trace.ddbj.nig.ac.jp/DRASearch/submission?acc=DRA003884 | Publicly available at the DDBJ Center (accession no. DRA00 |

Narushima S, Suda W, Imaoka A, Setoyama H, Nagamori T

3884)

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
