## [Decision Letter]

Thank you for submitting your article "Computational prediction of immune effectors from the microbiome to select for optimal immune–modulating consortia" for consideration by *eLife*. Your article has been reviewed by three peer reviewers, and the evaluation has been overseen by a Reviewing Editor and Wendy Garrett as the Senior Editor. The reviewers have opted to remain anonymous.

The reviewers have discussed the reviews with one another and the Reviewing Editor has drafted this decision to help you prepare a revised submission.

Summary:

In this manuscript, the authors colonize germ–free mice with individual bacterial strains, and test for induction of T_regs_. They then use this information to construct a model for induction of T_regs_ by a complex microbial community, reducing the number of combinations that must be tested explicitly. Specifically, they predict strong– and weak–inducing consortia and provide experimental validation of fit to the model. The authors use existing qPCR data from 13 strains and a generalized Lotka–Volterra model they previously published as the basis for the experimental validation. They deploy this model on 11–strain and 4–strain consortia. The three 11–strain combinations consist of two that lack strains with the highest "keystone" scores, and one that lacks the strain with the lowest score. They also test five 4–strain consortia, again collecting information on the dynamics. They also provide additional predictions relevant to ulcerative colitis.

In our view, this manuscript provides a fascinating example of validation of computational predictions relating specific microbial strains to immune response. However, we believe that although the work is impressive, additional validation needs to be done and that the last disease test case is less convincing than the rest of the work. We recommend that this be either removed or that experimental validation be provided.

Overall this is an impressive body of work and, as the authors note, point the way towards better rational design of microbial communities that alter the immune system.

Essential revisions:

1) Several of the figures are misleading or unreadable. Figure 3 shows observed vs predicted data and r^2^ in log space, but the results are likely to look much worse in linear space given the pattern of scatter, and this should be disclosed to readers rather than concealed.

2) Figure 4B is unreadable, and per–strain correlations should be summarized in a different way that is easier to understand.

3) Figure 5A is also unreadable and it is impossible to match up the stacked bar and area presentation; some other form of showing the data is needed.

4) Because the pathway data are normalized, Mann–Whitney U is not appropriate (in related tasks, its false discovery rate has been shown to be up to 95%) and appropriate statistics for differential abundance that take compositionality into account should be used.

5) The theoretical predictions for the consortia transferred into mice with UC patient stool, while intriguing, should ideally be backed by experiment. The results showing model fit to data for the 4–strain communities are impressive but much weaker than the rhetoric in the text implies: Spearman rho of –0.47 means that more than three quarters of the variance in the rank order of the data, let alone the quantitative data, is not explained by their explanatory variable. So, although the association is statistically significant, its power to predict results is low. This component should be removed.

6) The authors have only moderate support for their primary hypothesis: that immune response as a function of microbial consortia can be predicted via a model. In the GF mouse case, I see the key data testing this hypothesis in Figure 4E. While the stated correlation is significant, I'm concerned this statistic is being used inappropriately. Incorporating the replicates into the correlation seems ill–advised (doing so, for example, could let you compute a correlation involving only two groups provided each had enough replicates within). Rather, if the authors want to use the replicates, an ANOVA with post–hoc tests seems more appropriate. Perhaps even better, the authors could correlate the mean observed percentage of CD4^+^FOXP3^+^ to the mean predicted TrIS across the 5 groups. The trends look promising. But, this may be too small a number of groups to actually power such a correlation; hence, my overall concern that there's not enough data validating the model.

7) The manuscript needs to make clear when the authors are using new vs. previously published, and in silico vs. measured data. Examples include Figure 4, which analyses the in-silico results using a mixture of metabolomic predictions and experimental validation. It seems Figure 1 and Figure 2 are efforts to clear up that confusion. Unfortunately, they didn't work in my case; perhaps it’s worth trying to merge those figures into a single flow chart?

8) Some variables in Equations 1–3 aren't explicitly defined in the text (and don't seem to be in the Materials and methods section either), including i, α, β, ε, and n. These need to be defined.

9) How many strains were available and how many strains were actually selected? The numbers mentioned in the text are inconsistent in different parts. I provide only a few examples below, but this inconsistency is present throughout the text.

– Introduction: "a mixture of 17 human–derived Clostridia (ref 9)".

– Introduction: "our previous work identified on the order of 15 possible candidate strains to select from (ref 9)".

– Results section: "we gavaged 14 mice with one of three possible 11–strain subsets from the original 13 strains (Figure 2)".

– Figure 1: "12 therapeutically relevant Clostridia strains in the lumen".

– Figure 2: "13 strain cocktail".

Bottom line, the authors should provide a clear layout/table of the strains that were identified in ref 9, of the strains that were used in the paper, and of the combinations. At the moment, this part is very confusing, and the numbers do not match throughout the text.

10) How many GF mice were actually used, and why do the authors use IQI GF mice for some experiments, and C57BL/6 GF mice for others? The mouse genetic background was changed for no apparent reason.

Also Introduction: "we gavaged 14 mice with one of three possible 11–strain subsets from the original 13 strains (Figure 2)".

Materials and methods section: "the three mixtures of 11 strains (described above) were orally inoculated into five IQI germ–free adult mice each."

How many mice were actually gavages with the 11 strains? 14 or 5? Please clarify the experimental design.

---

## [Author Response]

Essential revisions:1) Several of the figures are misleading or unreadable. Figure 3 shows observed vs predicted data and r^2^ in log space, but the results are likely to look much worse in linear space given the pattern of scatter, and this should be disclosed to readers rather than concealed.

We edited Figure 3 by removing the old scatter plot (old panel B) and substituted it with one where we calculated the Spearman correlation between observations and predictions at different discrete time points in order to not violate the assumption of independence of observations for correlation tests. Calculation of Spearman’s correlation between predictions and observations follows our previous work (Stein et al., 2013). See also subsection “Generation of multimodal microbiome–T_reg_ data”.

2) Figure 4B is unreadable, and per–strain correlations should be summarized in a different way that is easier to understand.

We agree with the reviewer’s comment. Accordingly, we modified Figure 4B to only focus on Strain 27. We added a new supplementary figure (Figure 4—figure supplement 1) to display the same analysis for all other strains each in a separate panel.

3) Figure 5A is also unreadable and it is impossible to match up the stacked bar and area presentation; some other form of showing the data is needed

Following the suggestion from the editor we removed the UC biota analysis. Hence, Figure 5 is now removed from the manuscript.

4) Because the pathway data are normalized, Mann–Whitney U is not appropriate (in related tasks, its false discovery rate has been shown to be up to 95%) and appropriate statistics for differential abundance that take compositionality into account should be used.

We agree with the reviewer’s comment. Similarly to the response to comment #3, this analysis is not anymore included as the UC-biota related work has been removed from the manuscript.

5) The theoretical predictions for the consortia transferred into mice with UC patient stool, while intriguing, should ideally be backed by experiment. The results showing model fit to data for the 4–strain communities are impressive but much weaker than the rhetoric in the text implies: Spearman rho of –0.47 means that more than three quarters of the variance in the rank order of the data, let alone the quantitative data, is not explained by their explanatory variable. So, although the association is statistically significant, its power to predict results is low. This component should be removed.

As suggested by the editor, we removed the data and analysis pertaining the UC biota scenario. Regarding the analysis of the 4–strain validation experiments see the response to comment #6.

6) The authors have only moderate support for their primary hypothesis: that immune response as a function of microbial consortia can be predicted via a model. In the GF mouse case, I see the key data testing this hypothesis in Figure 4E. While the stated correlation is significant, I'm concerned this statistic is being used inappropriately. Incorporating the replicates into the correlation seems ill–advised (doing so, for example, could let you compute a correlation involving only two groups provided each had enough replicates within). Rather, if the authors want to use the replicates, an ANOVA with post–hoc tests seems more appropriate. Perhaps even better, the authors could correlate the mean observed percentage of CD4^+^FOXP3^+^ to the mean predicted TrIS across the 5 groups. The trends look promising. But, this may be too small a number of groups to actually power such a correlation; hence, my overall concern that there's not enough data validating the model.

We fully agree with the reviewer’s comment and followed the reviewer’s suggestion to correlate the mean CD4^+^FOXP3^+^ Treg level for each consortium with the modeling-predicted Treg induction score. Pearson’s correlation yields a coefficient of 0.97 and a p-value < 0.01. This should sufficiently validate our modeling predictions. We accordingly modified the text (see subsection “Derivation of the Treg Induction Score (TrIS) and selection of T_reg_-inducing consortia”), the related figure panel and the corresponding caption.

7) The manuscript needs to make clear when the authors are using new vs. previously published, and in silico vs. measured data. Examples include Figure 4, which analyses the in-silico results using a mixture of metabolomic predictions and experimental validation. It seems Figure 1 and Figure 2 are efforts to clear up that confusion. Unfortunately, they didn't work in my case; perhaps it’s worth trying to merge those figures into a single flow chart?

We agree with the reviewer and apologize for the confusion. We edited Figure 1 to be solely a clear conceptual diagram detailing the entire workflow that we present in the paper. We also rearranged Figure 2 and explicitly state what is new data and what data was published already. Also, we now explicitly specify (in Figure 2 and caption) what data was experimentally measured and what was derived from in silico simulations. We also better describe this in the main text (see Introduction and Results section). See also response to comment #9.

8) Some variables in Equations 1–3 aren't explicitly defined in the text (and don't seem to be in the Materials and methods section either), including i, α, β, ε, and n. These need to be defined.

Following the reviewer’s suggestion, we included a detailed description of the used parameters when they are introduced (see subsection “Derivation of the microbiome–T_reg_ mathematical model to select microbial consortia” and Discussion section).

9) How many strains were available and how many strains were actually selected? The numbers mentioned in the text are inconsistent in different parts. I provide only a few examples below, but this inconsistency is present throughout the text.– Introduction: "a mixture of 17 human–derived Clostridia (ref 9)".– Introduction: "our previous work identified on the order of 15 possible candidate strains to select from (ref 9)".– Results section: "we gavaged 14 mice with one of three possible 11–strain subsets from the original 13 strains (Figure 2)".– Figure 1: "12 therapeutically relevant Clostridia strains in the lumen".– Figure 2: "13 strain cocktail".Bottom line, the authors should provide a clear layout/table of the strains that were identified in ref 9, of the strains that were used in the paper, and of the combinations. At the moment, this part is very confusing, and the numbers do not match throughout the text.

We agree with the reviewer comment and again apologize for the confusion. Following the reviewers comment we made a table (Table 1) detailing membership for all the experiments performed for this work as well as from our previous studies. We also modified the text to clarify this point (Introduction).

10) How many GF mice were actually used, and why do the authors use IQI GF mice for some experiments, and C57BL/6 GF mice for others? The mouse genetic background was changed for no apparent reason.Also Introduction: "we gavaged 14 mice with one of three possible 11–strain subsets from the original 13 strains (Figure 2)".Materials and methods section: "the three mixtures of 11 strains (described above) were orally inoculated into five IQI germ–free adult mice each."How many mice were actually gavages with the 11 strains? 14 or 5? Please clarify the experimental design.

We made consistent and clarified in throughout text the number of animals used. The use of a C57BL/6 background for these experiments was motivated by animal availability at the time of performing experimental validation. C57BL/6 use can be justified by the fact that previous work from us has shown that Treg induction by our Clostridia strains does not differ between Balb/c, IQI, and C57BL/6 mice (see Atarashi et al., 2015). We reworded the text and this statement and added a paragraph to the Discussion section.